# Fitness, strength and severity of COVID-19: a prospective register study of 1 559 187 Swedish conscripts

Agnes af Geijerstam ![ORCID],[1] Kirsten Mehlig,[1] Mats Börjesson,[2,3] Josefina Robertson,[1,4] Jenny Nyberg ![ORCID],[5] Martin Adiels,[1,2] Annika Rosengren,[2,6] Maria Åberg,[1,7] Lauren Lissner[1]

For numbered affiliations see end of article.

**Correspondence to**
Dr Agnes af Geijerstam;
agnes.af.geijerstam@gu.se

## ABSTRACT

**Objective** To investigate the possible connection between cardiorespiratory fitness (CRF) and muscle strength in early adulthood and severity of COVID-19 later in life.

**Design** Prospective registry-based cohort study.

**Participants** 1 559 187 Swedish men, undergoing military conscription between 1968 and 2005 at a mean age of 18.3 (SD 0.73) years.

**Main outcome measures** Hospitalisation, intensive care or death due to COVID-19 from March to September 2020, in relation to CRF and muscle strength.

**Results** High CRF in late adolescence and early adulthood had a protective association with severe COVID-19 later in life with OR (95% CI) 0.76 (0.67 to 0.85) for hospitalisation (n=2 006), 0.61 (0.48 to 0.78) for intensive care (n=445) and 0.56 (0.37 to 0.85) for mortality (n=149), compared with the lowest category of CRF. The association remains unchanged when controlled for body mass index (BMI), blood pressure, chronic diseases and parental education level at baseline, and incident cardiovascular disease before 2020. Moreover, lower muscle strength in late adolescence showed a linear association with a higher risk of all three outcomes when controlled for BMI and height.

**Conclusions** Physical fitness at a young age is associated with severity of COVID-19 many years later. This underscores the necessity to increase the general physical fitness of the population to offer protection against future viral pandemics.

## BACKGROUND

The COVID-19 pandemic has affected the world in unprecedented ways. While measures such as contact restrictions and vaccinations relieve the immediate impact of the disease, epidemiological studies may identify risk factors that could be managed in the long run, in order to reduce the impact of future epidemics. Early on, advanced age and current cardiovascular comorbidities including obesity were found to be associated with a more severe course of COVID-19.[1–3]

Cardiorespiratory fitness (CRF) is regarded as one of the most important risk factors for overall health, cardiovascular morbidity and overall mortality.[4] The level of physical

### Strengths and limitations of this study

► Data from the Swedish military service conscription registry provided us with objective measures of fitness in a uniquely large sample.
► The prospective design with long and near-complete follow-up from well-validated hospital records and the death registry gives a strong ground for drawing conclusions.
► Limitations include thatit was not possible to distinguish conscripts with low versus extremely low fitness levels.
► Since female conscription was voluntary, our results can only be generalized to men.

activity early in life (and associated CRF) may have short-term and long-term effects on risk factors and the immune system. There has been speculation about the way it may confer protection against severe COVID-19.[5] An early observation from the UK Biobank cohort that slower self-reported walking pace predicted more severe COVID-19 suggests the importance of CRF for this particular disease,[6] and more recent studies have reinforced this notion, using self-reported measures[7 8] but also clinically tested CRF and strength.[9 10]

Although CRF is a persistent trait with a genetic component,[11] it is also to a large extent modifiable by regular exercise. Overall, physical fitness consists of CRF and muscular fitness, including muscle strength and healthy weight. Body mass index (BMI) is routinely measured in hospital and public care settings and has already been shown to be associated with severity of disease.[2 12] However, objective measures of CRF and strength before the onset of COVID-19 are rarely available in population-based studies. Self-reported measures are unreliable, and most studies conducted in middle-aged or older populations have limited information on risk factors in earlier life.[13] Participation rates among

young men in population-based studies also tend to be low.[14]

Swedish military service conscription registry contains detailed and high-quality information about CRF and strength in a uniquely large sample of young men. In contrast to many population-based surveys, the participation rates until the early 2000s suggested highly generalisable results. Combined with measured data on BMI and other potential confounding factors in early adulthood, linkage data from nationwide disease registries enable a life course perspective on the role of physical fitness as a potentially mitigating factor for severe COVID-19 that is presently lacking in the literature.

## METHODS
### Study design
This prospective cohort study is based on data from the Swedish military service conscription registry, the Longitudinal integration database for health insurance and labour market studies (LISA), and the Swedish national hospital, intensive care and cause of death registries (for description in detail, see Åberg *et al*).[15] Fitness data were collected in early adulthood, in relation to incidence data on COVID-19 retrieved in September 2020, after the first wave of the pandemic in Sweden.

### Patient and public involvement statement
As this is a registry-based study, there has been no patient or public involvement.

### Population studied
The Swedish military service conscription registry contains information about 1 949 891 Swedish individuals who enlisted for military service between late 1968 and 2005 giving between 15 and 52 years of follow-up until the outbreak of COVID-19. During that time, Swedish law required all male citizens to enlist, except for those in prison or those with severe chronic somatic or psychiatric conditions or functional disabilities (approximately 2%–3% annually). Due to a shift to voluntary recruitment in 2005, the data in later years is no longer considered representative. The standardised protocol included measurements of weight and height, blood pressure, muscular strength and CRF. The registry also includes prior medical diagnoses. The examinations took place over 2 days in six conscription centres across Sweden.

### Main independent variables
#### CRF score
To evaluate CRF, the subjects performed a cycle ergometric test where the work rate was successively increased until limited by exhaustion. Based on the final work rate, a nine-level score derived by the National Service Administration in Sweden[16] served as a standardised measure of CRF. These scores were validated in relation to different tasks the conscripts were expected to manage during their time in the military (cross-country running with a

heavy backpack and heavy lifting).[16] Over the decades, there have been some changes in the conscription examination protocols, and raw data were not available for all conscripts.[17] However, the CRF scores matching certain military requirements were assigned during all recruitment years and can be assumed to be comparable across years.[11] For this study, the scores were collapsed into three categories describing low,[1–5] medium[6 7] and high[8 9] CRF. The asymmetrical assignment of CRF scores was prompted by the fact that the fitness data in the lowest categories were omitted from the registry after 1999.[15]

### Muscle strength score
Isometric muscle strength in units of newton was assessed as a weighted sum of knee extension (weight 1.3), elbow flexion (weight 0.8) and handgrip (weight 1.7). As the methods for measuring muscle strength did not change over the years and both the weighted values and assigned strength scores were available in the registry until 1996, it was possible to examine strength as a continuous variable.[16]

## OTHER COVARIATES
### BMI, height and weight status
Weight and height were measured by standard anthropometric measurement techniques, and continuous BMI values ($kg/m^2$) were calculated and also divided into categories of underweight (BMI<18.5), normal weight (BMI 18.5–24.9), overweight (BMI 25.0–29.9), obesity (BMI 30–34.9) and obesity class 2 (BMI≥35).[18] For those aged 15–17 years at conscription, weight status categories were adjusted based on age-specific and sex-specific BMI z-scores, according to the WHO recommended childhood BMI cutoffs.[19] In addition, height was included as a covariate, given its positive association with muscular strength.

### Blood pressure
High blood pressure in early adulthood was used as an indicator of predisposition to later hypertension and other cardiovascular diseases (CVDs).[6] Systolic and diastolic blood pressure was measured according to a standardised protocol after 5–10 min of rest in the supine position. In this study, values were then categorised into five subgroups (optimal blood pressure (systolic blood pressure <120 mm Hg and diastolic blood pressure <80 mm Hg), normal blood pressure (120–129 and 80–84 mm Hg), high normal blood pressure (130–139 and 85–89 mm Hg), grade 1 hypertension (140–159 and 90–99 mm Hg) and grade 2 hypertension (≥160 and ≥100 mm Hg) based on the 2018 European guidelines.[20]

### Chronic disease at baseline
To assess the specific effect of physical fitness, we controlled for morbidity at baseline, using International Classification of Diseases (ICD) codes for respiratory disease

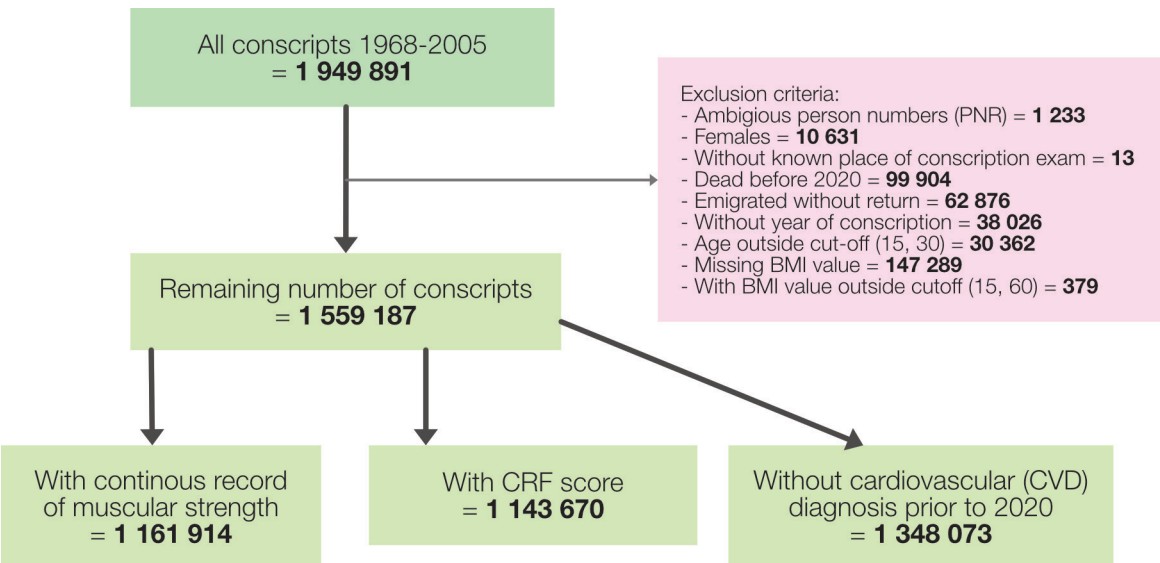

**Figure 1** Construction of analytic sample. BMI, body mass index; CRF, cardiorespiratory fitness.

(J00–J99), CVD (I00–I99), diabetes (E10-14), kidney disease (N00-N29) and malignant cancers (C00-C97).

### Parental education

Parental education was considered a proxy for socioeconomic position of the family of origin. Using data from the LISA registry, parental education was based on the highest of maternal and paternal education and divided into three categories with low education (up to 9 years), medium education (high school diploma with ≤2 years at university) and high (≥3 years at university).[17]

### Analytic samples and subsamples

Figure 1 describes the construction of the analytic sample. Among 1 949 891 individuals examined for conscription between 1968 and 2005, the following exclusions were made: those with ambiguous identification numbers or without known place or year of conscription, those dying or emigrating without return before 2020 and all female conscripts, leaving 1 737 217 observations. Because of the aim of relating fitness measured in adolescence and early adulthood to disease severity later in life, individuals with age at conscription outside the interval of 15–30 years were excluded. Since BMI was an important confounder for the association between fitness and disease, the sample was further restricted excluding all men with missing or implausible BMI values (range for inclusion set between 15 and 60). Among the remaining 1 559 187 men, 1 143 670 had values for CRF and 1 161 914 had registered values for the continuous measure of muscle strength.

### Outcome parameters: hospitalisation and/or death due to COVID-19

Using the Swedish personal identification number, the full sample was linked to the national hospital, the intensive care and the cause of death registries. From these, all cases between March and September 2020 with a main diagnosis of ICD U071 for test verified infection with

SARS-CoV-2 and U072 for clinically diagnosed COVID-19, or deaths with U071 or U072 as an underlying cause of death were identified.

Records with U071 or U072 as a secondary diagnosis were counted as cases if the main diagnosis was clinically related to COVID-19 (full list in online supplemental table S1). This resulted in a reduction of total cases from 2957 to 2635.

All cases were considered as severe COVID-19 since they required hospital care. Furthermore, patients were categorised into three groups with increasing severity:

► Hospitalisation due to COVID-19: n=2594.
► Intensive care due to COVID-19: n=559.
► Deaths due to COVID-19: n=184.

Register data based on Swedish hospital records have high validity.[21] As the healthcare system was stressed during 2020 but never overloaded, it is unlikely that a significant number of potential hospital patients with COVID-19 were not admitted or included in the records.

### STATISTICAL ANALYSIS

The main independent variable, CRF, was analysed in terms of three categories describing low, medium and high values of exposure. Logistic regression was used to calculate the odds for COVID-19 outcome by this exposure category, adjusted for regional examination centre, age at and year of conscription, as well as linear, quadratic and cubic terms of BMI, height, parental education, chronic disease and blood pressure at examination. Due to the rarity of events, in particular deaths from COVID-19, we used a penalised likelihood estimation (Firth method) to reduce possible small sample bias in maximum likelihood estimation.[22]

Effect modification was investigated by including product terms between the main exposure (CRF categories) and other categorical predictors into the model,

overweight status (BMI ≥25 kg/m$^2$), parental education, blood pressure category and current age dichotomised at median value (52 years). For each interaction analysis, the p value from an F-test comparing a model with and without interactions was assessed. Results were given in terms of ORs and 95% CIs. Muscle strength in units of newton (N) was standardised to zero mean and unit SD, and results were given in terms of OR per SD. To explore the non-linear association between muscle strength and COVID-19, we used logistic regression based on restricted cubic splines.[23] Considering the rarity of events, the ORs may be interpreted as relative risk, according to the rare disease assumption.

To investigate whether the associations with CRF or strength were explained by differences in CVD) during midlife, we conducted a mediation analysis including confounding of exposure–mediator–outcome associations with Delta method for CIs.[24] A supplementary analysis about the highest levels of fitness was also conducted.

Statistical analyses were performed with SAS V.9.4 (SAS Institute). Statistical significance was set at 0.05 (two-sided tests).

### Ethics approval statement
The study conforms to the principles outlined in the Declaration of Helsinki. The data were anonymised before being accessed by the study authors.

## RESULTS
Basic characteristics of the 1 559 187 men included in the study are shown by conscription decade in table 1. The majority of deaths and hospitalisations due to COVID-19 were observed among those who underwent conscription between 1968 and 1985.

### CRF in early adulthood and severity of COVID-19
Our analysis shows a protective association between higher CRF at conscription and hospitalisation, intensive care or death due to COVID-19 later in life, adjusted for examination year and place and conscript's age at examination. ORs for high versus low fitness were 0.76 (95% CI 0.67 to 0.85), 0.61 (95% CI 0.48 to 0.78) and 0.56 (95% CI 0.37 to 0.85) respectively.

This association remains statistically significant with only minor attenuation after also controlling for BMI (figure 2, left panel), as well as blood pressure, height, baseline morbidity and parental education level (full results in online supplemental table S2).

In order to examine whether there was any additional or decreased protection at the highest level of CRF, categories 8 and 9 were disaggregated, but no statistically significant difference could be seen between the two subgroups. Adjusting for muscle strength did not change the results for CRF.

Interaction analyses indicated that the associations between CRF and all three COVID-19 endpoints were not modified by weight status, blood pressure, baseline morbidity or education. For example, comparing high versus low fitness in overweight (BMI group 3–5) and non-overweight conscripts (BMI group 1–2), the association with intensive care requiring COVID-19 tended to be stronger in the overweight group, OR=0.49 (0.25–0.96), than in the non-overweight group, OR=0.65 (0.50–0.85), but not significantly so (p value for interaction=0.5). Similarly, the other three tested interactions were not statistically significant (p>0.1).

### Muscle strength and severity of COVID-19
Muscular strength, a continuous variable measured in units of newton and recorded until 1997, showed associations with COVID-19 that were similar in direction to the more robust protective association with CRF. As shown in figure 2, right panel, when analysed in intervals of 313 N (=1 SD), the association was statistically significant for all three outcomes, but after controlling for cardiovascular fitness, the results were attenuated (full results in online supplemental table S3). Further analysis using cubic spline analyses confirmed the association between muscle strength and COVID-19. Specifically, these analyses, illustrated in supplementary materials (online supplemental figure S1), confirmed that lower muscle strength in young adulthood has a significant linear association for all three endpoints and no significant non-linear component.

### Supplementary mediation analysis
A supplementary analysis of CVD as a mediating factor in the association between CRF or muscle strength and severe COVID-19 was conducted. This analysis indicated that the direct effects of CRF and muscular strength (not mediated by CVD) are only slightly weaker than the overall total effect. Full results of this analysis are available in online supplemental table S4.

## DISCUSSION
The main finding of this study is that lower CRF and lower muscle strength in young adulthood are predictive of severe illness and death due to COVID-19 15–52 years later. The protective association between cardiovascular fitness and COVID-19 severe enough to warrant hospitalisation, intensive care and/or death was not attenuated by controlling for BMI at the time of conscription examinations. It was also independent of parental education, chronic morbidity and blood pressure at conscription.

As of this date, no other comparable study has been published on the link between fitness in young adulthood and later COVID-19. Using data from the Swedish military conscript registry provided us with objective measures of fitness in a uniquely large sample. This together with the prospective design, the long and near-complete follow-up from well-validated hospital records and the death registry provides a strong ground for drawing conclusions.

Other studies point in the same direction, though with a shorter time span between exposure and outcome. Measured hand grip strength has been shown to correlate

**Table 1** Description of analytical sample by conscription decade

| Conscription year | 1968–1975 | 1976–1985 | 1986–1995 | 1996–2005 | All decades |
|---|---|---|---|---|---|
| N | 283 082 | 441 414 | 496 484 | 338 207 | 1 559 187 |
| Age at conscription, mean (SD) | 18.5 (0.65) | 18.3 (0.82) | 18.3 (0.76) | 18.2 (0.58) | 18.3 (0.73) |
| Age in 2020, mean (SD) | 65.9 (1.94) | 57.5 (2.98) | 48.0 (3.00) | 38.3 (2.91) | 51.8 (9.89) |
| **CRF** | | | | | |
| n with CRF score | 248 424 | 380 791 | 358 731 | 155 720 | 1 143 670 |
| CRF low, n (%) | 89 953 (36.2) | 135 569 (35.6) | 85 203 (23.8) | 25 546 (16.4) | 336 271 (29.4) |
| CRF medium, n (%) | 84 189 (33.9) | 127 965 (33.6) | 189 000 (52.7) | 87 343 (56.1) | 488 497 (42.7) |
| CRF high, n (%) | 74 286 (29.9) | 117 257 (30.8) | 84 528 (23.6) | 42 831 (27.5) | 318 902 (27.9) |
| **Muscle strenght** | | | | | |
| n with strength score in newton | 277 150 | 435 323 | 449 188 | n/a | 1 161 661 |
| Mean score (SD) | 2 033 (299) | 2 092 (315) | 2 140 (315) | n/a | 2 096 (313) |
| **BMI** | | | | | |
| Mean (SD) | 21.2 (2.6) | 21.7 (2.8) | 22.0 (3.0) | 22.7 (3.6) | 21.9 (3.0) |
| Overweight and obese, n (%) | 20 481 (1.5) | 44 265 (2.2) | 65 268 (3.2) | 64 943 (5.6) | 194 957 (3.6) |
| **Height** | | | | | |
| Mean cm (SD) | 178.7 (6.4) | 179.1 (6.5) | 179.3 (6.6) | 179.8 (6.7) | 179.2 (6.7) |
| **Blood pressure** | | | | | |
| Grade 1 hypertension, n (%) | 51 242 (18.1) | 81 990 (18.6) | 89 752 (18.5) | 57 582 (21.4) | 280 566 (19) |
| Grade 2 hypertension, n (%) | 2 167 (0.8) | 3 145 (0.7) | 2 685 (0.6) | 1 614 (0.6) | 9 611 (0.7) |
| **Baseline morbidity** | | | | | |
| n with CVD (%) | 8 025 (2.8) | 10 836 (2.5) | 13 836 (2.8) | 4 732 (1.4) | 37 429 (2.4) |
| n with diabetes (%) | 102 (0.04) | 255 (0.06) | 321 (0.06) | 273 (0.08) | 951 (0.06) |
| n with kidney disease (%) | 285 (0.1) | 399 (0.09) | 508 (0.1) | 316 (0.09) | 1 508 (0.1) |
| n with malignant cancer (%) | 71 (0.03) | 134 (0.03) | 371 (0.07) | 231 (0.07) | 807 (0.05) |
| n with respiratory disease (%) | 22 760 (8.04) | 49 808 (11.3) | 88 904 (17.9) | 66 155 (19.6) | 227 627 (14.6) |
| **Parental education** | | | | | |
| Low, n (%) | 101 918 (52.9) | 141 205 (34.8) | 85 586 (17.5) | 28 186 (8.4) | 356 895 (25) |
| Medium, n (%) | 77 020 (40) | 213 050 (52.5) | 309 246 (63.1) | 222 095 (65.8) | 821 411 (57.6) |
| High, n (%) | 13 787 (7.2) | 51 660 (12.7) | 95 589 (19.5) | 87 069 (25.8) | 248 105 (17.4) |
| **COVID-19 diagnoses** | | | | | |
| n in hospital due to COVID-19 (%) | 728 (0.26) | 944 (0.21) | 674 (0.14) | 248 (0.07) | 2 594 (0.17) |
| n in intensive care due to COVID-19 (%) | 195 (0.07) | 209 (0.05) | 130 (0.03) | 25 (0.01) | 559 (0.04) |
| n dead due to COVID-19 (%) | 117 (0.04) | 43 (0.01) | 24 (0.00) | 0 | 184 (0.01) |

CRF, cardiorespiratory fitness.

with COVID-19 hospitalisations among adults 50 years and older.[10] Also being consistently inactive is strongly associated with increased risk of severe COVID-19.[7]

A possible mechanism for associations between CRF and severe COVID-19 could be higher cardiovascular morbidity (diagnosed or not) in the years after conscription examination among those with low CRF. Earlier studies using the same data have shown an association between low fitness and higher risk of heart failure, diabetes type 2, stroke, hypertension, ischaemic heart disease and psychiatric disorders.[15 25–28] In this regard, CRF could also be a marker for a better long-term lifestyle (less smoking, better diet).

Alternatively, there could be a direct anti-inflammatory effect or a modulation of the immune system, either acquired during adolescence and early adulthood and retained during the years leading up to 2020 or continually reinforced by physical activity established earlier in life.[29] In a subsample of 139 609 individuals, we had access to more recent CRF data collected from a health profiling registry,[30] showing a clear correlation between these estimates of $VO2_{max}$ and the fitness categories from conscription, indicating that physical fitness tracks over time.

However, only a minor part of the protective association between CRF and COVID-19 outcome was mediated by

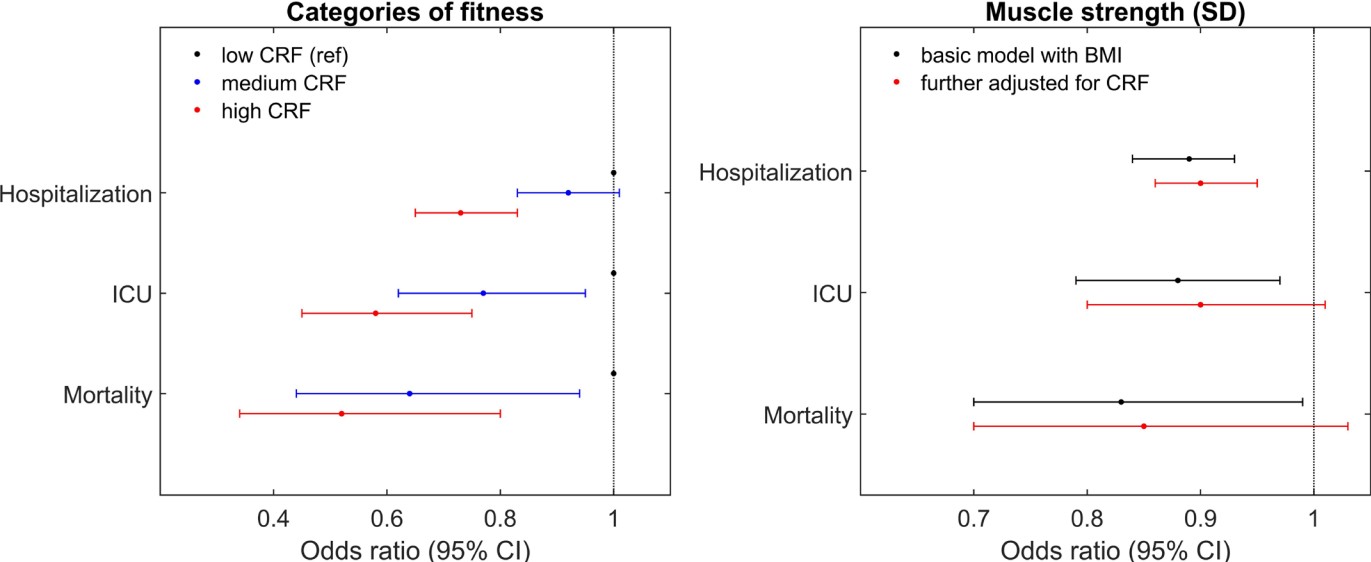

**Figure 2** Association between categories of cardiorespiratory fitness (CRF) and severe COVID-19 (left panel, n=1 143 670, hospitalisations=2006, intensive care=445 and deaths=149) and between muscle strength as a continuous predictor and severe COVID-19 (right panel, or per SD of muscle strength, n=1 161 827, hospitalisations=2252, intensive care=514 and deaths=180). All regression models were made with Firth's bias correction and adjusted for year and place, as well as age and BMI at conscription examination. BMI, body mass index.

protection from CVD in midlife. This points to a genuine effect of physical fitness acting via a modulation of the immune system. Of course, we cannot rule out a genetic predisposition to lower inflammatory levels, in conjunction with genetically higher CRF.[31]

Despite a number of strengths, the study is not without limitations, including the fact that a large number of Swedish citizens have immigrated as adults and never went through the conscription examination. Immigrants have been over-represented in the Swedish COVID-19 mortality and morbidity statistics with explanations ranging from crowded living conditions, lack of official information and occupations where working from home is not an option. This group has a lower socioeconomic status and higher degree of lifestyle-related risk factors (more obesity, more smoking and more diabetes). Importantly, this group has also lower CRF, potentially contributing to explaining their higher risk.[32]

Moreover, the results from this cohort can only be generalised to men, since the conscription process was not mandatory for women. Other limitations include some protocol changes in strength and CRF measurements over the years. For instance, a rather large number of conscripts are lacking CRF scores, predominantly during the last decade where the lowest scores were not recorded. This makes it difficult to draw conclusions regarding very low CRF levels and could underestimate the adverse association. In contrast to the CRF scores, the muscular strength measurements among the conscripts are normally distributed, and the analysis shows a considerable increase in the odds of developing severe COVID-19 among those at the lowest end of the distribution. When CRF was included in the model, the effect of mus, indicating that CRF is the more important factor.

Height turned out to be independently predictive of hospitalisation with OR 1.014 (1.006–1.021) per cm (95% CI, basic model adjusted for BMI), with a possible explanation being the association between height and venous thromboembolism.[33]

A related concern is the current age range of the cohort (32–70 years in 2020). All reported findings were adjusted for exact age at conscription and year of conscription examination and thus indirectly for age in 2020. To further illustrate this point, we repeated the analysis after stratification at median current age (52 years) and found only minor differences between the age groups. This underscores that protective associations of higher fitness are observed independent of cohort differences in current age or length of follow-up.

The present study has clear clinical implications. Physical fitness is a (largely) modifiable exposure and is therefore an important preventive measure both affecting later CVD and, as this study shows, severe COVID-19. The obvious public health implications ought to be to further strengthen efforts concerning fitness in the young and presumably through the life course. Anecdotal stories about severely sick athletes during 2020 (the first patient in intensive care in Italy was a marathon runner) raised the question if extremely physically fit individuals had a higher risk of severe COVID-19 but no statistically significant difference could be seen between those of very high CRF (score 9) and those with moderately-high (score 8), although the power of this analysis was low.

Looking towards the future, there are also the indirect effects of the pandemic and subsequent lockdowns on physical activity behaviours. In Sweden, a country with relatively few restrictions during the pandemic, only small changes in lifestyle habits have been shown so far,[34]

even though the amount of time sitting down is similar to other study settings.[35 36]

Considering the fact that women seem to run lower risk for severe COVID-19,[37] it is unfortunate not to be able to compare the protective effects of CRF in women and men. Using other sources of data such as the Swedish medical birth register, this might be done in a future study. Finally, although we found the results for CRF were generally robust and unaffected by adjusting for covariates such as BMI, we cannot exclude the possibility of residual confounding by unmeasured factors. This includes later exposures and comorbidities occurring between conscription and 2020.

## CONCLUSION

There is growing evidence of the importance of CRF on morbidity later in life,[30] whereas this study shows fitness at a young age may influence the severity level of COVID-19 many years later. In the near future, updates of the COVID-19 data will make it possible to understand these effects in more detail but are unlikely to reverse the results.

The findings of the present study reinforce the need to promote regular physical activity early in life to increase the general CRF and muscular strength of the population to decrease the risk of future cardiovascular events and other conditions and to offer protection against potential consequences of future viral pandemics.

**Author affiliations**
¹School of Public Health and Community Medicine, Institute of Medicine, University of Gothenburg Sahlgrenska Academy, Goteborg, Sweden
²Department of Molecular and Clinical Medicine, University of Gothenburg Sahlgrenska Academy, Goteborg, Sweden
³Center for Health and Performance, University of Gothenburg, Goteborg, Västra Götaland, Sweden
⁴Department of Infectious Diseases, Institute of Biomedicine, University of Gothenburg Sahlgrenska Academy, Goteborg, Sweden
⁵Section for Clinical Neuroscience, Institute of Neuroscience and Physiology, University of Gothenburg Sahlgrenska Academy, Goteborg, Sweden
⁶Sahlgrenska University Hospital, Goteborg, Sweden
⁷Regionhälsan, Region Västra Götaland, Göteborg, Sweden

**Contributors** LL and MÅ initiated the project. AaG and KM performed all statistical analyses. AaG had main responsibility for writing the article. MB, JR, JN, MA and AR all made substantial contributions to the interpretation of the analyses, the structure and content of the manuscript and have read and approved of the final draft. All authors have agreed to be accountable for all aspects of the work.

**Funding** This work was supported by the EpiLife-Teens Research Program (FORMAS2012-00038), the Swedish ALF-agreement (ALFGBG-720201) and the Swedish Research Council (02508, VRREG 2019-00193, 2020-05792).

**Disclaimer** The funding sources had no role in study design, collection, analysis and interpretation of data, the writing of the report, or the decision to submit the article for publication.

**Competing interests** None declared.

**Patient consent for publication** Not required.

**Ethics approval** The Ethics Committee of the University of Gothenburg and Confidentiality Clearance at Statistics Sweden approved the study (EPN Reference numbers EPN 462-14 and 567-15; T174-15, T653-17, T196-17, T 2020-01325,

T 2020-02420). The requirement for informed consent was waived by the Ethics Committee of the University of Gothenburg for secondary analysis of existing data.

**Provenance and peer review** Not commissioned; externally peer reviewed.

**Data availability statement** Data may be obtained from a third party and are not publicly available. The data used in this study is available on request from the Swedish National Board of Health and Welfare, the Swedish intensive care registry and Statistics Sweden.

**ORCID iDs**
Agnes af Geijerstam http://orcid.org/0000-0002-0897-6548
Jenny Nyberg http://orcid.org/0000-0002-4336-3886

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
