## [Reviewer comments · BMJ Open]

ARTICLE DETAILS

TITLE (PROVISIONAL)	Fitness, strength and severity of COVID-19 – a prospective register study on 1 559 187 Swedish conscripts.
AUTHORS	af Geijerstam, Agnes; Mehlig, Kirsten; Börjesson, Mats; Robertson, Josefina; Nyberg, Jenny; Adiels, Martin; Rosengren, Annika; Åberg, Maria; Lissner, Lauren

VERSION 1 – REVIEW

REVIEWER	Cheval, Boris University of Geneva
REVIEW RETURNED	05-Apr-2021

GENERAL COMMENTS	General comments This study aimed to investigate the association of cardiorespiratory fitness and muscle strength (assessed in early adulthood) with severe COVID-19. Using a cohort study of 1 559 187 Swedish men, results showed that higher cardiorespiratory fitness and higher muscle strength was associated with lower odds of COVID-19 hospitalization, hospitalization in intensive care unit and mortality. The authors concluded that physical fitness is associated with COVID-19 severity, stressing the need to improve general physical fitness of the population. The paper is well-written and easy to follow. The topic is relevant for BMJ open as it uncovers the role of physical fitness in reducing the risk of hospitalization or death due to severe COVID-19. This study includes at least four features that limit the conclusions that can be drawn. First, the models were not adjusted for several established risk factors for severe COVID-19 including underlying conditions such as respiratory diseases, kidney diseases, diabetes and cancers. Because these comorbidities are associated physical fitness, the models cannot estimate the associations between physical fitness and COVID-19 severity over above the influence of thee established risk factors for severe COVID-19. In other words, the authors cannot assess the independent (or unique) association between physical fitness and severe COVID-19. Second, participants completed the physical fitness test during their early adulthood. As such, there is a large temporal gap in between the measurement of the “exposure” and of the primary outcome. This is problematic as the exposure is expected to evolve over the lifespan and, most importantly, this evolution is likely to be marked by large individual variabilities. Accordingly, the early life physical fitness is a rather biased indicator of the physical fitness at the time
---

of the potential SARS-CoV-2 infection. This may have biased the associations observed. Of note, this bias can either decrease or reinforce the strength of the associations observed.

Third, the study only involved men. As such, it is not possible to determine whether the associations observed also apply in women or if they were specific to men (this limitation was already acknowledged).

Fourth, the events are rare – 2594 hospitalizations on 1 559 187 participants, i.e., ~0.17% of the sample; 559 intensive care, i.e., 0.04%; and 184 deaths, i.e., 0.01%. In such situation, using rare events logistic regression (at least in supplementary analyses) seems appropriate (see example here: http://docs.zeligproject.org/articles/zelig_relogit.html).

Because the data are correlational, I would avoid the use of terms that may induce causal links between the variables (e.g., protective effect).

Additional relevant literature linking physical fitness and COVID-19 severity could have been mentioned, such as:
Brawner CA, Ehrman JK, Bole S et al. Maximal exercise capacity is inversely related to hospitalization secondary to coronavirus disease 2019. *Mayo Clin Proc.* 2021;96(1):32-9.

Cheval, B., Sieber, S., Maltagliati, S., Millet, G.P., Formánek, T., Chalabaev, A., Cullati, S. and Boisgontier, M.P., 2021. Muscle strength is associated with COVID-19 hospitalization in adults 50 years of age and older. *MedRxiv*.

Maltagliati, S., Sieber, S., Sarrazin, P., Cullati, S., Chalabaev, A., Millet, G.P., Boisgontier, M.P. and Cheval, B., 2021. Muscle Strength Explains the Protective Effect of Physical Activity against COVID-19 Hospitalization among Adults aged 50 Years and Older. *medRxiv*.

Salgado-Aranda, R., Pérez-Castellano, N., Núñez-Gil, I., Orozco, A.J., Torres-Esquivel, N., Flores-Soler, J., Chamaisse-Akari, A., McInerney, A., Vergara-Uzcategui, C., Wang, L. and González-Ferrer, J.J., 2021. Influence of Baseline Physical Activity as a Modifying Factor on COVID-19 Mortality: A Single-Center, Retrospective Study. *Infectious Diseases and Therapy*, pp.1-14.

Moreover, it would have been interesting to discuss that data showing that usual physical activity behaviors (a key predictor of physical fitness) have been largely disturbed during the pandemic period. This will reinforce the importance of ensuring sufficient level of physical activity during the pandemic.

Cheval, B., Sivaramakrishnan, H., Maltagliati, S., Fessler, L., Forestier, C., Sarrazin, P., ... & Boisgontier, M. P. (2020). Relationships between changes in self-reported physical activity, sedentary behaviour and health during the coronavirus (COVID-19) pandemic in France and Switzerland. *Journal of sports sciences*, 1-6.

Dunton, G.F., Wang, S.D., Do, B. and Courtney, J., 2020. Early effects of the COVID-19 pandemic on physical activity locations and

behaviors in adults living in the United States. Preventive Medicine Reports, 20, p.101241.

Rhodes, R.E., Liu, S., Lithopoulos, A., Zhang, C.Q. and Garcia-Barrera, M.A., 2020. Correlates of Perceived Physical Activity Transitions during the COVID-19 Pandemic among Canadian Adults. Applied Psychology: Health and Well-Being, 12(4), pp.1157-1182.

Overall, I believe that the paper has some merits and is a strong candidate for publication in BMJ open.

Specific comments:

Methods

- Regarding the main exposures (i.e., cardiorespiratory fitness and muscle strength). Is that possible to have more information on the quality of the tests performed, and therefore their validity?
- If muscle strength is used as a continuous variable, height (and not only BMI) should be accounted for (height have a direct influence of muscle strength, irrespective of individuals' physical health).
- Categorizing continuous predictors has been criticized (<https://pubmed.ncbi.nlm.nih.gov/16217841/>)
- Of note, it is surprising to treat one variable in continuous (muscle strength) and the other one as categorical variable. Moreover, this choice is difficult to understand because clinical cut-offs have been developed and validated for muscle strength.
- Regarding the main outcomes, it should be discussed the reliability of the information obtained (i.e., discuss whether all the cases of COVID-19 hospitalization or death has been identified). I do not know the case of Sweden, but in some European countries, the identification of COVID-19 infection was problematic. Likewise, in some European countries, because of the pressure on the health system, people with severe case of COVID-19 was not always hospitalized.
- As aforementioned, regarding the statistical analysis, rare events logistic regressions could be tested.
- Regarding the sensitivity analysis, a mediation analysis seems more appropriate to assess whether cardiovascular diseases explained the association between physical fitness and severe COVID-19.

Results

- P10, L43-45: It is interesting to observe that cardiovascular disease seems to explain a part of the link between muscle strength and severe COVID-19. The percentage of effect reduction could be added here.

Discussion

- Comparison of the results with other studies is rather lacking.
- A discussion about the absence of adjustment for established risk factors for severe COVID-19 is lacking (see general comments).
- Similarly, authors should discuss how the temporal gap between physical fitness measurement and COVID-19 hospitalization may have biased the associations observed.

Thank you for giving me the opportunity to read this work. I hope that my comments will be useful for the authors.

REVIEWER	Oliveira Souto, Fabrício Universidade Federal de Pernambuco, Nucleo de Ciencias da Vida
REVIEW RETURNED	07-Apr-2021

GENERAL COMMENTS	The study by Geijerstam et al., points to evidence that fitness at a young age may influence the severity level of COVID-19, even after many year later. The study is very innovative and adds to the recent works that has been published relating physical activity and the severity of COVID-19. In addition, although it is not addressed directly in this study, reinforces the need to promote regular physical activities to improve the immune response against viral infections, for example. In general, the study is well written. The abstract is adequate and structured, following a logical sense. The analysis of the work presents a good approach, bringing a combination of a retrospective and prospective analysis, leading to a prospective registry-based cohort study. The results, as well as the conclusion, are in line with the objectives proposed in the study. Some points:  -On page 5, lines 26-27, the authors mention " BMI is routinely measured in hospital and public care settings and has already been shown to be associated with severity of disease(1)", I did not find the information that the BMI is associated with the severity of COVID, please check the reference. - On page 6, line 5, the authors describe that 1938027 Swedish individuals were registered from 1968 to 2005. However, on page 7, line 26, the value is 1949891. Are these the same? - Figure 2, page 18, describes "All regression models were adjusted for year and place, as well as age and BMI at conscription examination". To make the association between categories of cardiorespiratory fitness (CRF) and severe COVID-19, was the association made using the total number of individuals in the study compared with the categories of the COVID-19 diagnosis? Or, were only the patients by category of diagnosis used, compared with their previous data for CRF? As a suggestion, the number of patients for each analysis or category could be included in the figure caption. Minor points:  - Uniform "COVID-19", there are two forms in the text "Covid-19" and "COVID-19". - Please review the text and references. Suggestion for future analysis: Although the authors do not describe in the study, a good tool to investigate the rate of physical activity in a large population is the International Physical Activity Questionnaire (IPAQ). This can be suggested to be applied in the population hospitalized with non-severe COVID-19 or for those without the need for intubation.
---

VERSION 1 – AUTHOR RESPONSE

Reviewer: 1
Dr. Boris Cheval, University of Geneva
Comments to the Author:

Paper: Fitness and strength in young Swedish men predicted incidence of severe COVID-19 during the first wave of the pandemic – a prospective register study.

Date: 31 March 2021

General comments

This study aimed to investigate the association of cardiorespiratory fitness and muscle strength (assessed in early adulthood) with severe COVID-19. Using a cohort study of 1 559 187 Swedish men, results showed that higher cardiorespiratory fitness and higher muscle strength was associated with lower odds of COVID-19 hospitalization, hospitalization in intensive care unit and mortality. The authors concluded that physical fitness is associated with COVID-19 severity, stressing the need to improve general physical fitness of the population.

The paper is well-written and easy to follow. The topic is relevant for BMJ open as it uncovers the role of physical fitness in reducing the risk of hospitalization or death due to severe COVID-19.

This study includes at least four features that limit the conclusions that can be drawn.

1. First, the models were not adjusted for several established risk factors for severe COVID-19 including underlying conditions such as respiratory diseases, kidney diseases, diabetes and cancers. Because these comorbidities are associated physical fitness, the models cannot estimate the associations between physical fitness and COVID-19 severity over above the influence of these established risk factors for severe COVID-19. In other words, the authors cannot assess the independent (or unique) association between physical fitness and severe COVID-19.

Response:

We originally adjusted the model for baseline blood pressure and BMI. But as this comment makes clear, we need to adjust for other chronic diseases at conscription in order to extract the original (unconfounded) effect of physical activity in adolescence and early adulthood. Baseline respiratory disease, kidney disease, CVD, diabetes and cancer are now included in the model, the results remained unchanged. This is added to the covariates and results sections.

2. Second, participants completed the physical fitness test during their early adulthood. As such, there is a large temporal gap in between the measurement of the “exposure” and of the primary outcome. This is problematic as the exposure is expected to evolve over the lifespan and, most importantly, this evolution is likely to be marked by large individual variabilities. Accordingly, the early life physical fitness is a rather biased indicator of the physical fitness at the time of the potential SARS-CoV-2 infection. This may have biased the associations observed. Of note, this bias can either decrease or reinforce the strength of the associations observed.

Response:

We would like to stress that the main aim of this study was the identification of early risk factors for later COVID-19 severity. Information on fitness in young adulthood is not usually available. But we thank you for raising this issue as it prompted us to do an additional analysis of the fitness data considering how CRF tracks over time. A subsample of this cohort was followed for physical fitness after their conscription examinations (in this case a sub-maximal ergometer cycling test) in conjunction with health profiling of employees of a large number of Swedish companies. There is clearly a correlation between this data on VO₂max and the earlier fitness categories from conscription. Also it shows that the overall fitness decreases linearly with age and the time between conscription and the later exam. This may be a topic for an upcoming publication.

3. Third, the study only involved men. As such, it is not possible to determine whether the associations observed also apply in women or if they were specific to men (this limitation was already acknowledged).

Response:

Yes, we agree that this is a limitation. We still decided to use conscript data because it is a large and representative set of data with comprehensive information on physical fitness early in life. In Brawner et al they stratified between the sexes, but the trend for lower likelihood of hospitalization in relation to maximal exercise capacity in women was not significant. Hopefully later studies will answer this question more thoroughly.

4. Fourth, the events are rare – 2594 hospitalizations on 1 559 187 participants, i.e., ~0.17% of the sample; 559 intensive care, i.e., 0.04%; and 184 deaths, i.e., 0.01%. In such situation, using rare events logistic regression (at least in supplementary analyses) seems appropriate (see example here: http://docs.zeligproject.org/articles/zelig_relogit.html).

Response:

Thank you for the suggestion, Firth's bias correction is now included in the main analysis.

5. Because the data are correlational, I would avoid the use of terms that may induce causal links between the variables (e.g., protective effect).

Response:

Thank you, this is corrected.

6. Additional relevant literature linking physical fitness and COVID-19 severity could have been mentioned, such as:

Response:

Thank you for the suggestions, more recent papers are incorporated.

Brawner CA, Ehrman JK, Bole S et al. Maximal exercise capacity is inversely related to hospitalization secondary to coronavirus disease 2019. *Mayo Clin Proc.* 2021;96(1):32-9.

Cheval, B., Sieber, S., Maltagliati, S., Millet, G.P., Formánek, T., Chalabaev, A., Cullati, S. and Boisgontier, M.P., 2021. Muscle strength is associated with COVID-19 hospitalization in adults 50 years of age and older. *MedRxiv*.

Maltagliati, S., Sieber, S., Sarrazin, P., Cullati, S., Chalabaev, A., Millet, G.P., Boisgontier, M.P. and Cheval, B., 2021. Muscle Strength Explains the Protective Effect of Physical Activity against COVID-19 Hospitalization among Adults aged 50 Years and Older. *medRxiv*.

Salgado-Aranda, R., Pérez-Castellano, N., Núñez-Gil, I., Orozco, A.J., Torres-Esquivel, N., Flores-Soler, J., Chamaisse-Akari, A., McInerney, A., Vergara-Uzcategui, C., Wang, L. and González-Ferrer, J.J., 2021. Influence of Baseline Physical Activity as a Modifying Factor on COVID-19 Mortality: A Single-Center, Retrospective Study. *Infectious Diseases and Therapy*, pp.1-14.

7. Moreover, it would have been interesting to discuss that data showing that usual physical activity behaviours (a key predictor of physical fitness) have been largely disturbed during the pandemic period. This will reinforce the importance of ensuring sufficient level of physical activity during the pandemic.

Response:

This is a very interesting point for further research. In Sweden, a country with relatively few restrictions during the pandemic, only small changes in lifestyle habits have been shown so far (Blom et al. ref #34). We have added a section on this in the discussion, thank you for the paper suggestions.

Cheval, B., Sivaramakrishnan, H., Maltagliati, S., Fessler, L., Forestier, C., Sarrazin, P., ... & Boisgontier, M. P. (2020). Relationships between changes in self-reported physical activity, sedentary

behaviour and health during the coronavirus (COVID-19) pandemic in France and Switzerland. Journal of sports sciences, 1-6.

Dunton, G.F., Wang, S.D., Do, B. and Courtney, J., 2020. Early effects of the COVID-19 pandemic on physical activity locations and behaviors in adults living in the United States. Preventive Medicine Reports, 20, p.101241.

Rhodes, R.E., Liu, S., Lithopoulos, A., Zhang, C.Q. and Garcia-Barrera, M.A., 2020. Correlates of Perceived Physical Activity Transitions during the COVID-19 Pandemic among Canadian Adults. Applied Psychology: Health and Well-Being, 12(4), pp.1157-1182.

Overall, I believe that the paper has some merits and is a strong candidate for publication in BMJ open.

Specific comments:

Methods

8. Regarding the main exposures (i.e., cardiorespiratory fitness and muscle strength). Is that possible to have more information on the quality of the tests performed, and therefore their validity?

Response:

The standard reference validating the Conscript's registry methods was published in 1974 but is not readily available, (Nordesjö, L. O., and R. Schéle, ref #15).

In this publication, the authors validated the ergometer cycle test and the measures of isometric muscle strength in relation to 1) 2,8 km cross country running with a backpack weighting 22 kg; 2) lifting of 20 kg 100 times; 3) carrying of 17 kg in each hand for as long as possible without rest. They could show high correlation between the cycle test and the running criterion, and a lower but significant correlation between the strength test and the lifting and carrying criteria.

This information is now summarized in the revised manuscript (methods section). If of further interest, we can provide this article scanned as a pdf.

9. If muscle strength is used as a continuous variable, height (and not only BMI) should be accounted for (height have a direct influence of muscle strength, irrespective of individuals' physical health).

Response:

Height is correlated with strength and is therefore now adjusted for. Mention of this has been incorporated into the model descriptions. It also turned out that height was independently predictive of hospitalization due to COVID-19, as mentioned in the discussion section.

10. Categorizing continuous predictors has been criticized

(<https://pubmed.ncbi.nlm.nih.gov/16217841/>)

Response:

We agree and thank you for the reference. The data were not collected for our particular research question but for classification of conscripts' potential to conduct specific physical tasks. The changes in examination protocol over the years are a cause for concern. In addition, raw data on fitness were not available for all conscripts, as now emphasized in limitations.

For this reason, we used the variables that are most comparable across years, e.g. categories of CRF. Because methods for muscle strength did not change over time, analyses could be based on the original continuous variable.

11. Of note, it is surprising to treat one variable in continuous (muscle strength) and the other one as categorical variable. Moreover, this choice is difficult to understand because clinical cut-offs have been developed and validated for muscle strength.

Response:

Please see response to the question above.

12. Regarding the main outcomes, it should be discussed the reliability of the information obtained (i.e., discuss whether all the cases of COVID-19 hospitalization or death has been identified). I do not know the case of Sweden, but in some European countries, the identification of COVID-19 infection was problematic. Likewise, in some European countries, because of the pressure on the health system, people with severe case of COVID-19 was not always hospitalized.

Response:

This is a good point, and we further comment on it under outcomes. Hospital records in Sweden have high validity (Ludvigsson JF et al. Ref # 21), but this is of course not including COVID-19.

13. As aforementioned, regarding the statistical analysis, rare events logistic regressions could be tested.

Response:

Please see our answer to question 4.

14. Regarding the sensitivity analysis, a mediation analysis seems more appropriate to assess whether cardiovascular diseases explained the association between physical fitness and severe COVID-19.

Response:

Thank you for this comment, a mediation analysis has been performed. Mediation accounted for 4.8% of the CRF association. Results are summarized in the paper and additional details are presented in supplementary material. The conclusions were consistent with the previous analysis.

Results

15. P10, L43-45: It is interesting to observe that cardiovascular disease seems to explain a part of the link between muscle strength and severe COVID-19. The percentage of effect reduction could be added here.

Response:

Good point, we also conducted a mediation analysis for strength. The proportion mediated is now reported to be 4.0% for strength, compared to 7.4 for CRF (hospitalization). This is added to the results.

Discussion

16. Comparison of the results with other studies is rather lacking.

Response:

Some more literature has been added, thank you for your suggestions.

17. A discussion about the absence of adjustment for established risk factors for severe COVID-19 is lacking (see general comments).

Response:

As mentioned earlier, this is added to the text.

18. Similarly, authors should discuss how the temporal gap between physical fitness measurement and COVID-19 hospitalization may have biased the associations observed.

Response:

Please see our answer to question 2.

Thank you for giving me the opportunity to read this work. I hope that my comments will be useful for the authors.

Reviewer: 2

Dr. Fabrício Oliveira Souto, Universidade Federal de Pernambuco

Comments to the Author:

The study by Geijerstam et al., points to evidence that fitness at a young age may influence the severity level of COVID-19, even after many years later. The study is very innovative and adds to the recent works that have been published relating physical activity and the severity of COVID-19. In addition, although it is not addressed directly in this study, reinforces the need to promote regular physical activities to improve the immune response against viral infections, for example.

In general, the study is well written. The abstract is adequate and structured, following a logical sense. The analysis of the work presents a good approach, bringing a combination of a retrospective and prospective analysis, leading to a prospective registry-based cohort study. The results, as well as the conclusion, are in line with the objectives proposed in the study.

Some points:

19. On page 5, lines 26-27, the authors mention " BMI is routinely measured in hospital and public care settings and has already been shown to be associated with severity of disease(1)", I did not find the information that the BMI is associated with the severity of COVID, please check the reference.

Response:

Thank you, the reference is updated.

20. On page 6, line 5, the authors describe that 1938027 Swedish individuals were registered from 1968 to 2005. However, on page 7, line 26, the value is 1949891. Are these the same?

Response:

Thank you, this is a typo. The correct number is 1 949 891.

21. Figure 2, page 18, describes "All regression models were adjusted for year and place, as well as age and BMI at conscription examination". To make the association between categories of cardiorespiratory fitness (CRF) and severe COVID-19, was the association made using the total number of individuals in the study compared with the categories of the COVID-19 diagnosis? Or, were only the patients by category of diagnosis used, compared with their previous data for CRF? As a suggestion, the number of patients for each analysis or category could be included in the figure caption.

Response:

Thank you, the numbers are included. Those in the COVID-19 categories were compared to the total number of individuals in the study population.

Minor points:

22. Uniform "COVID-19", there are two forms in the text "Covid-19" and "COVID-19".

- Please review the text and references.

Response:

Thank you, this is corrected.

Suggestion for future analysis:

Although the authors do not describe in the study, a good tool to investigate the rate of physical activity in a large population is the International Physical Activity Questionnaire (IPAQ). This can be suggested to be applied in the population hospitalized with non-severe COVID-19 or for those without the need for intubation.

Response:

Thank you for this suggestion, if there is a follow-up case-control study, IPAQ would be our method of choice!

VERSION 2 – REVIEW

REVIEWER	Cheval, Boris University of Geneva
REVIEW RETURNED	27-May-2021

GENERAL COMMENTS	After reading the revision and the response letter, I consider the authors were overall responsive to my comments on the previous version of the manuscript. Of note the Cheval et al.'s paper is now published in Journal of Cachexia, Sarcopenia, and Muscle. This reference can therefore be updated. I have no further comment, and I am therefore happy to support the publication of this paper Congratulations to the authors
--